# Large-Scale, High-Resolution Mapping of Soil Aggregate Stability in Croplands Using APEX Hyperspectral Imagery

**Pu Shi [1,2,*]**, **Fabio Castaldi [3]**, **Bas van Wesemael [2]** and **Kristof Van Oost [2]**

1   College of Earth Sciences, Jilin University, Changchun 130061, China
2   Georges Lemaître Centre for Earth and Climate Research, Earth and Life Institute, Universite Catholique de Louvain, 1348 Louvain-la-Neuve, Belgium; bas.vanwesemael@uclouvain.be (B.v.W.); kristof.vanoost@uclouvain.be (K.V.O.)
3   ILVO—Flanders Research Institute for Agriculture, Fisheries and Food, Technology and Food Science-Agricultural Engineering, 9820 Merelbeke, Belgium; fabio.castaldi@ilvo.vlaanderen.be
*   Correspondence: shipu@jlu.edu.cn or pu.shi@uclouvain.be

**Abstract:** Investigations into the spatial dynamics of soil aggregate stability (AS) are urgently needed to better target areas that have undergone soil degradation. However, due to the lack of efficient alternatives to the conventional labor-intensive methods to quantify AS, detailed information on its spatial structure across scales are scarce. The objective of this study was to explore the possibility of using hyperspectral remote sensing imagery to rapidly produce a high-resolution AS map at regional scale. Airborne Prism Experiment (APEX) hyperspectral images covering an area of 230 km$^2$ in the Belgian loam belt were used together with a local topsoil dataset. Partial least squares regression (PLSR) models were developed for three AS indexes (i.e., mean weight diameter (MWD), microaggregate and macroaggregate fractions) and soil organic carbon (SOC), and evaluated against an independent validation dataset. The prediction models were then applied to more than 700 bare soil fields for the production of high resolution (2×2 m) MWD and SOC maps. The PLSR models had a satisfactory level of accuracy for all four variables (R$^2$ >0.5, RPD > 1.4), and the predicted maps were capable of capturing the fine-scale as well as the between-field variabilities of soil properties. Variogram analysis on the spatial structure of MWD showed a clear spatial organization at the catchment scale (range: 1.3 km) that is possibly driven by erosion-induced soil redistribution processes. Further analysis in restricted areas displayed contrasting spatial structures where spatial auto-correlation of AS was only found at field scale, thus highlighting the potential of hyperspectral remote sensing as a promising technique to investigate the spatial variability of AS across multiple scales.

**Keywords:** soil aggregate stability; APEX; hyperspectral imagery; spatial variability; soil erosion

## 1. Introduction

Soil aggregate stability (AS) is controlled by an array of elementary soil properties such as soil organic carbon (SOC), texture and extractable metal oxides [1,2]. Agricultural management practices and topographic positions also influence its dynamics, thus making AS a dynamic property that evolves with space and time [3,4]. Decrease of AS in croplands not only hinders agricultural production through the control over surface crusting and seedling emergence [5], but also increases the risks of soil degradation, thereby compromising the physical protection of SOC by intact soil structure [6] and amplifying the erosion-induced nutrient and pollutant transfer from soils to surface water [7]. Among all influencing factors, SOC is frequently reported to be an essential one that positively controls the dynamics of AS [8–10]. Meersmans et al. [11] reported a decreasing trend in cropland SOC content

during a 50-year period, which increased the risk for soil structural degradation, especially in areas where SOC concentrations were close to 1% or lower. The Voluntary Guidelines for Sustainable Soil Management published by the Food and Agriculture Organization of the United Nations [12] also identified the increase of SOC stock as an important measure to combat soil degradation. In light of the current sustainable soil management efforts aiming to mitigate climate change (e.g. "4 per 1000" initiative [13]), numerous studies have shown the positive impact of crop residue retention and conservation tillage on AS in croplands [10,14,15]. To better understand the underlying mechanisms leading to the dynamics of SOC and AS, and to target areas that are vulnerable to soil degradation and nutrient losses, it is necessary to monitor SOC and AS over large spatial and long temporal scales.

The importance of accounting for the spatiotemporal variability of AS is also recognized in soil erosion assessments. As an indicator of soil erodibility to reflect a soil's inherent resistance to external erosive forces [16], AS is often used as an input into spatially distributed soil erosion models to investigate erosion-induced soil redistribution patterns [17]. Performance assessments of such models pointed to the gap between satisfactory spatial predictions of soil erosion and insufficient representation of the spatial variability of soil erodibility [18]. Due to the lack of large-scale, high-resolution AS data, current modelling practices often assume soil erodibility (as indicated by AS) as a static parameter, instead of treating it as a spatiotemporally dynamic input. Among limited number of studies that assessed the spatial variability of AS, Mohanmmadi and Motaghian [19] showed the large spatial variation of AS with a range value of ca. 3 km from variogram analysis; Annabi et al. [20] also demonstrated the spatial structure of AS in a geologically diverse region and pointed out the need to capture small-scale variability of AS by increasing the sampling density. However, with wet-sieving remaining to be the conventional method to measure AS in those studies, it is unrealistic to analyze a large number of samples required to assess the large-scale variability of AS at a fine spatial resolution. This highlights the necessity to develop new methods that allow efficient quantification of AS at large scales and fine resolution.

Recent developments of hyperspectral remote sensing techniques have shown the potential to map key soil properties, such as topsoil SOC and soil texture [21], both of which are important determinants of AS. In particular, the Airborne Prism Experiment (APEX) [22] sensor offers high spectral and spatial resolution images and has been proven to be capable of predicting SOC at field to catchment scales [23]. Other airborne hyperspectral imaging applications in digital soil mapping include developing a spectral index to correction for soil moisture effects [24], and analysis of spatial organizations of mapped soil properties (e.g. $CaCO_3$, iron and cation exchange capacity) [25]. Apart from using hyperspectral images to predict and characterize soil properties, spatial patterns of erosion and deposition could also be characterized by developing classification methods with spectrally-predicted elementary soil properties as inputs [26], and by matching soil properties of different soil horizons emerging at the surface with different soil erosion and deposition stages [27].

As of yet, few studies used hyperspectral remote sensing images to directly map secondary soil physical properties across large scales, while laboratory-based hyperspectral data have already been extensively explored to predict properties, such as AS, that are related to known soil chromophores [28]. For instance, soil mean weight diameter (MWD), a lumped index commonly used to express AS, and different aggregate size fractions were successfully predicted using laboratory visible-infrared (Vis-NIR) spectroscopy [29]. The authors attributed the good model performance to the close correlation between AS and SOC, as the wavelengths that are known contributors to the prediction of SOC were also found to be significant in the MWD prediction. This warrants further investigations on whether the successful application of hyperspectral imagery to SOC mapping could be transferred to the mapping of AS.

The objective of this study was to develop a method for large-scale, high-resolution AS mapping for the investigation into the spatial dynamics of AS. To this end, we aim to test the capability of APEX hyperspectral imagery to predict AS across an agricultural region in Belgium at a 2×2 m spatial resolution. The approach used in this study began with extracting bare soil fields based on pre-defined spectral indexes that are representative of bare soils. Then, partial least squares regression (PLSR)

models were established using the hyperspectral data extracted from APEX images. Finally, the AS prediction model was evaluated against an independent validation dataset and an AS map of the study area was produced. Such a map will not only provide detailed information on field-level soil degradation status for the sake of precision agriculture, but at the same time allow assessments of spatial variation of AS at multiple scales.

## 2. Materials and Methods

### 2.1. Study Region

The study region, over which APEX hyperspectral images were acquired in 2013, 2015 and 2018, is located in the center of the Belgian loam belt (Figure 1). The area is a ca. 230 km² strip (SW corner: 50.59N, 4.69E; NE corner: 50.70N, 5.11E) from Gembloux to Lincent. It is characterized by geological formations of the Quaternary and Tertiary Eras, with loess-derived haplic Luvisols (IUSS Working Group WRB, 2015) as the major soil type. The climate in this region is temperate oceanic with mean temperatures between 2.3 °C (January) and 17.8 °C (July), and the mean annual precipitation is 790 mm, which is evenly distributed throughout the year [30]. Cropland is the dominant land use type with productive silt loam soils on a rolling topography. Main crops are winter wheat, winter barley, sugar beet, maize and potatoes. SOC content in this region is decreasing due to the intensive agriculture, making it increasingly vulnerable to soil degradation [11].

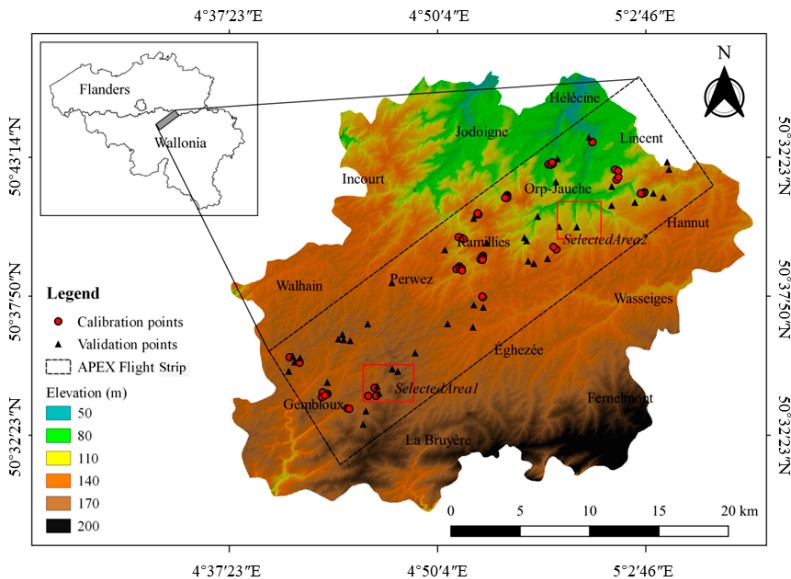

**Figure 1.** Location of the study region over which the APEX hyperspectral images (dashed rectangle) were acquired, and calibration and validation samples were collected. The red rectangles are the two representative areas selected for the spatial analysis of aggregate stability.

### 2.2. APEX Hyperspectral Images

The Airborne Prism Experiment, referred to as APEX, is a Belgian-Swiss consortium development on behalf of the European Space Agency. It is a dispersive push broom imaging spectrometer, that is mounted on a Leica PAV-30 stabilizing platform in the Dornier DO-228 aircraft (German Aerospace Center) with N2 pressure system, covering the wavelength range from 400 to 2500 nm [22]. Several flight campaigns have been completed since its onset in 2010, covering more than 160 areas across Europe [31]. In this study, the APEX flight organized on September 2, 2018, covering a large percentage of bare fields, was used for the prediction of AS. To enable the examination on the quality of the APEX hyperspectral data, ground truth Vis-NIR reflectance spectra were collected at designated locations on

the same day of the APEX flight, using an ASD Fieldspec 3 FR spectroradiometer (Analytical Spectral Devices Inc., Boulder, CO, USA).

The APEX hyperspectral data were pre-processed by the Central Data Processing Center (CDPC) at the VITO Remote Sensing Department, Mol, Belgium. Radiometric, spectral, and geometric calibrations were carried out with the calibration cubes generated from data collected on the APEX Calibration Home Base (CHB) at DLR, Oberpfaffenhofen, Germany [32]. Atmospheric correction was conducted with the MODTRAN4 radiative transfer model following the algorithms given in [33]. Geometric correction was performed by means of direct georeferencing using a C++ module developed by VITO [34]. Input data from the sensor's GPS/IMU, together with the boresight correction data and the ASTER DEM were used during the ortho-rectification process. Then, the post-processed images were resampled to a spatial resolution of $2 \times 2$ m and projected to WGS 84/UTM zone 31N. Each data cube contains 285 spectral bands. Water absorption bands at 1339.5–1417.6 nm and 1795.4–1949.3 nm regions and bands on the edge of spectra influenced by noise were removed, resulting in 255 bands for the subsequent analysis.

*2.3. Local Soil Dataset*

A soil sampling campaign was completed in October 2018, approximately one month after the acquisition of APEX images. 83 topsoil (0–10 cm) samples were collected, and at each sampling location, a composite sample comprising five subsamples taken within a 3 m radius was prepared. Then, the fresh samples were air-dried and divided into two subsets for soil analyses. The first subset was sieved through a 2 mm mesh and analyzed for laboratory Vis-NIR spectra, SOC, texture, and pH, while the second subset was passed through 3 mm and 5 mm sieves and soil aggregates between 3 and 5 mm were stored at room temperature for AS analysis. In particular, laboratory Vis-NIR spectra were obtained with an ASD Fieldspec 3 FR spectroradiometer (Analytical Spectral Devices Inc.), as detailed in [29]. Total carbon concentration was measured by dry combustion with a VarioMax CN analyzer (Elementar GmbH, Langenselbold, Germany), and for the samples showing clear reactions under 10% HCl treatment, inorganic carbon content was measured using a modified pressure-calcimeter method [35]. Then, SOC was obtained by subtracting the inorganic carbon content from total carbon. Soil texture was measured by laser diffraction (LS 13320, Beckman Coulter, Brea, California, USA) after removing organic matter with 35% $H_2O_2$, and pH was measured at 1:2.5 soil/water ratio by a pH meter (PHS-3E, Leici, Shanghai, China). On average, the collected soils had a SOC concentration of 1.3%, pH of 6.8, and clay, sand and silt fractions of 10.6%, 68.6% and 20.8% respectively.

AS was measured following the fast wetting treatment proposed in [8]. Briefly, 7–8 g of 3–5 mm soil aggregates were subjected to fast wetting in deionized water for 10 minutes, in order to disintegrate the aggregates through slaking. Next, the fragments were sieved through a 63 μm sieve in ethanol. The fragments larger than 63 μm were dried in the oven at 40 °C for 48 h, and sieved through a column of sieves to get the mass percentages for 63–125, 125–250, 250–500, 500–1000, and 1000–2000μm size classes. Finally, aggregate mean weight diameter (MWD) was calculated by multiplying the mean diameter of each size class with the mass percentage of the respective size class and dividing the sum of these products from all size classes by their total mass percentages. Besides MWD, mass percentages for microaggregates (63–250 μm) and macroaggregates (250–2000 μm) were aggregated and used as response variables in the prediction of AS.

*2.4. Development of Prediction Models for AS*

Figure 2 depicts the work flow of the development and evaluation of prediction models for AS. Detailed descriptions of each procedure are described below.

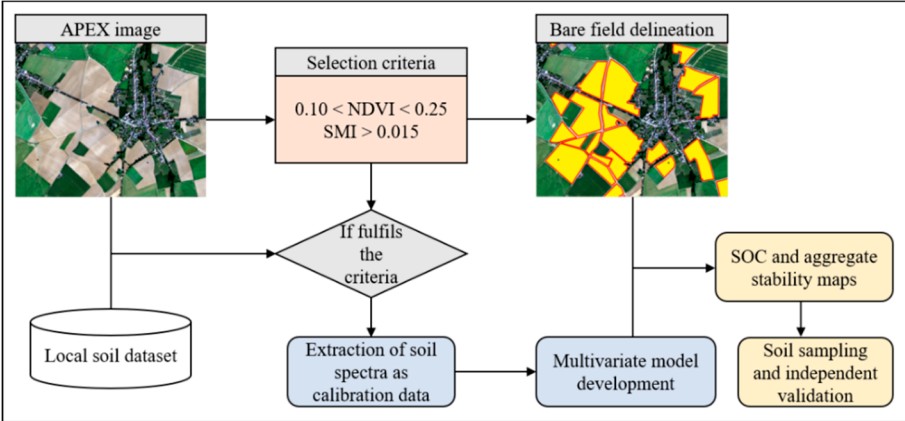

**Figure 2.** Flow chart of the APEX image processing, model development and evaluation procedure.

2.4.1. Bare Field Selection

In order to map soil properties, the first and foremost task is to extract bare soil fields that are free of vegetation and build-up areas from the images. It is well established that the normalized different vegetation index (NDVI) can be used to eliminate the green vegetation. Also, soil moisture index (SMI) was proven to be useful in minimizing the effect of soil moisture on soil spectra [24] and can be calculated as follows:

$$SMI = \frac{\rho_{2049} - \rho_{2193}}{\rho_{2049} + \rho_{2193}} \tag{1}$$

where the normalized difference between the reflectance at wavelength 2049 and 2193 nm was used to indicate the soil moisture level. A higher SMI value indicates a drier soil surface. A SMI threshold of 0.015 was adopted from Diek et al. [24] to indicate dry conditions. SMI values above this threshold were assumed not to be disturbed by the noise caused by variation in topsoil moisture. Next, to determine the meaningful threshold for NDVI that is most representative of the bare soils, 100 optimally-conditioned bare fields were visually assessed and selected from the APEX images using the RGB bands, and the NDVI range of these fields were used as thresholds. Consequently, the NDVI threshold was set between 0.10 and 0.22. Lastly, the two pre-defined spectral indexes were used as bare soil selection criteria and applied to the entire study area. A bare soil field was delineated if the entire field fulfilled the criteria while at the same time excluding road networks and built-up areas. Thus, 788 fields (41.2 km$^2$) were extracted from the raw APEX images, covering approximately 18% of the study area.

2.4.2. Development of Prediction Models for AS and SOC

To construct a calibration dataset for the development of prediction models, the same criteria used during the bare field selection were applied to the data points in the local soil dataset (see above). For the geolocations that were detected as bare, the corresponding APEX spectra were extracted using the "bilinear" method from the "extract" function provided by the R package "raster". Then, the calibration dataset was formed by combining the extracted APEX spectra and the analytical measurements of soil properties (i.e., SOC, MWD, microaggregate and macroaggregate fractions), where the bands of the spectra are the independent predictors, while the soil properties are the response variables. A principal component analysis was performed on the APEX spectra to calculate the standardized Mahalanobis distance (H) between each spectrum and the average spectrum [36]. Spectra with H > 3 were treated as outliers and were excluded from subsequent analysis. The reflectance spectra were then transformed to absorbance (i.e., log(1/Reflectance)) prior to the partial least squares regression (PLSR) to build prediction models for the AS indexes (i.e., MWD, microaggregate and macroaggregate fractions). It should be noted that a PLSR model was also built for SOC following the same approach, in order to compare its performance and influencing factors to those of MWD (see below). Lastly, the

calibrated models for SOC and MWD were applied to all bare soil pixels of the entire study region, and high-resolution SOC and MWD maps, covering 788 fields, were produced.

Tenfold cross-validation (CV) was performed to assess the predictive capability of the models. Coefficient of determination ($R^2$), root mean square error (RMSE), Ratio of Performance to Deviation (RPD) values were used to evaluate the performance of (cross) calibration. The Variance Importance Projection (VIP) index, a weighted sum of squares of the PLS weights, was calculated for both MWD and SOC estimations. Spectral bands with VIP values greater than one are considered significant variables for the PLSR model. Furthermore, the laboratory Vis-NIR spectra were resampled to the APEX spectral resolution and subjected to the same modelling procedure as that of the extracted APEX spectra. VIP indexes for both MWD and SOC estimations using the resampled laboratory ASD spectra were then compared to the VIP indexes resulting from APEX-based models.

### 2.4.3. Independent Validation Dataset

To further assess the robustness of the calibrated model, an independent validation dataset was created in October 2019. To avoid the time-consuming laboratory AS measurements, the independent validation was carried out against the SOC model. As previously reported [29], SOC and MWD had a close correlation in the study area and the spectral regions that contributed to the prediction of MWD were known SOC predictors, we thus assumed that the model validation result against SOC would largely translate into that against MWD prediction. A total of 43 topsoil samples were collected following a stratified random sampling strategy. The predicted SOC map was divided into three SOC concentration classes (i.e., <1%, 1–1.5%, and >1.5%), corresponding to increasing AS levels because of the positive correlation between the two variables. In each class, 13–15 samples were randomly selected at a one point per field density, in order to cover as many fields as possible. The same sampling procedure used for the calibration dataset was adopted for the validation dataset. All samples were prepared in the laboratory and measured for SOC content using the same protocol.

### 2.5. Analysis of the Spatial Variability of AS

Empirical variograms were estimated for MWD to allow for the examination of spatial structure of AS in the study region. Firstly, 300,000 pixels were selected across the entire region by random sampling and the empirical variogram was computed by the method-of-moments [37]. Secondly, in order to investigate the small-scale variation in AS and potential distinct spatial structures for different areas, two restricted areas (ca. 2 km²) that represent typical topographic and soil conditions of the study region were selected. Then, empirical variograms, each with 100,000 randomly selected pixels, were generated for these two areas. Theoretical variogram models (exponential, spherical or wave model) were fitted to empirical models using weighted least squares method. The range of these three variogram models were used to characterize the spatial variation of AS at multiple scales. All statistical analyses were carried out with R (version 3.5.1).

## 3. Results and Discussion

### 3.1. Summary of Soil Properties for the Calibration and Validation Datasets

Out of the 83 samples in the local soil dataset, 49 samples met the selection criteria for bare soil points and were thus included in the calibration dataset. The study region is characterized by a low SOC content, which led to generally low levels of AS (Table 1). According to the AS classes proposed by Le Bissonnais [8], a mean MWD of 0.35 mm represents highly unstable soils. This is reflected in a higher average percentage of microaggregates in comparison to that of macroaggregates, meaning that the soil aggregates were generally prone to breakdown forces and more fine fragments would be generated in the event of heavy rainfall impact. The mean SOC concentration and its standard deviation were similar for both the calibration and validation dataset, while the maximum SOC concentration in the calibration dataset was higher.

**Table 1.** Descriptive statistics of the SOC and aggregate stability indexes for the calibration and validation datasets.

| Variable [1] | Minimum | Maximum | Mean | Std |
|---|---|---|---|---|
| **Calibration (n = 49)** | | | | |
| SOC (g 100 g$^{-1}$) | 0.75 | 2.02 | 1.20 | 0.28 |
| MWD (mm) | 0.20 | 0.77 | 0.35 | 0.14 |
| Microaggregates (%) | 26.27 | 72.58 | 45.20 | 9.04 |
| Macroaggregates (%) | 18.34 | 60.63 | 35.93 | 10.94 |
| **Validation (n = 43)** | | | | |
| SOC | 0.76 | 1.64 | 1.18 | 0.25 |

[1] SOC, soil organic carbon; MWD, mean weight diameter; microaggregates and macroaggregates denote the mass percentages of in the 63–250 and 250–2000 μm size fractions.

Similar to what was reported in a previous study that used the entire local soil dataset to explore the relationship between SOC and MWD [29], a positive correlation (r = 0.79, P < 0.01) between the two variables was found in the calibration dataset. Moreover, there appeared to be a two-stage pattern in the relationship, with a SOC concentration at 1.5% likely being the critical threshold (Figure 3). For soils that have SOC concentrations higher than 1.5%, there was a clear linear relationship that shows the substantial increase in MWD with SOC, whereas the soils with lower SOC concentrations (<1.5%) mostly formed a cluster characterized by a flattened slope of the fitted line. Also, the MWD for samples with low SOC concentrations was mostly in the 0.2-0.4 mm range, and the correlation coefficient between the two variables decreased by more than two times. This implies that the continuing decrease in SOC content in the study region could cause a concurrent degradation of AS, until reaching a critical point (i.e., 1.5%) below which the soils become highly unstable (MWD < 0.4). This stresses the importance of conservation agriculture as a means to increase SOC concentration in the topsoil and thus in turn restore soil structural stability. It also highlights the necessity to develop remote sensing techniques as a tool to monitor the change in SOC and AS. Remote sensing techniques are a promising tool to monitor the change in SOC and AS, because soil degradation is a threat to soil surface properties in particular in croplands.

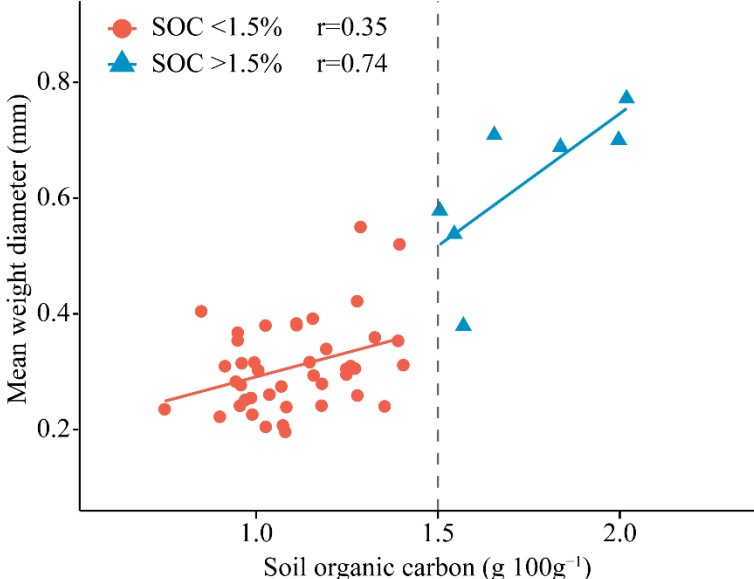

**Figure 3.** Scatterplot of soil mean weight diameter (MWD) versus soil organic carbon (SOC). We divided the data into two parts using a SOC threshold of 1.5 g 100 g$^{-1}$. Solid lines for both parts were fitted by linear regression.

### 3.2. SOC and AS Model Development and Evaluation

Comparison between the extracted APEX spectra and the ground truth ASD spectra showed similarities both in terms of the general pattern of the spectrum and the absolute reflectance (Figure 4a). There are some noisy wavelengths close to the two water absorption bands around 1400 nm and then around 1700−1800 nm, but the generated APEX spectra in general appeared to be well in line with the ground truth. Then, the APEX spectra were used to develop PLSR-based prediction models for SOC and three AS indexes (i.e., MWD, microaggregates and macroaggregates fraction). Two spectral outliers were identified, and the elimination of the outliers did not affect the statistical distribution of SOC and MWD. According to the cross-validation results (Table 2), all four prediction models had $RPD_{cv}$ values higher than 1.4 and $R^2_{cv}$ values higher than 0.5, while the highest $RPD_{cv}$ and $R^2_{cv}$ were achieved for the microaggregate fraction. The fact that satisfactory models were achieved for all four response variables, in particular the three AS indexes, proved the capability of APEX hyperspectral images to predict AS. An acceptable model for SOC was also achieved, with $RMSE_{cv}$ at 0.19 g C per 100 g soil. This is consistent with a previous study that used APEX data to map SOC in the same region, which had a similar level of RMSE (0.15 g 100 $g^{-1}$) and RPD (1.4) [23].

**Table 2.** Estimation accuracy of PLSR model calibration (cal) and cross-validation (cv) for SOC, MWD, microaggregates and macroaggregate fractions.

| Variable | $RMSE_{cal}$ g 100$g^{-1}$ | $R^2_{cal}$ | $RPD_{cal}$ | $RMSE_{cv}$ g 100$g^{-1}$ | $R^2_{cv}$ | $RPD_{cv}$ |
|---|---|---|---|---|---|---|
| SOC | 0.15 | 0.70 | 1.84 | 0.19 | 0.52 | 1.42 |
| MWD | 0.06 | 0.79 | 2.23 | 0.09 | 0.58 | 1.52 |
| Microaggregates | 4.10 | 0.79 | 2.22 | 5.68 | 0.61 | 1.60 |
| Macroaggregates | 4.97 | 0.78 | 2.17 | 6.85 | 0.60 | 1.58 |

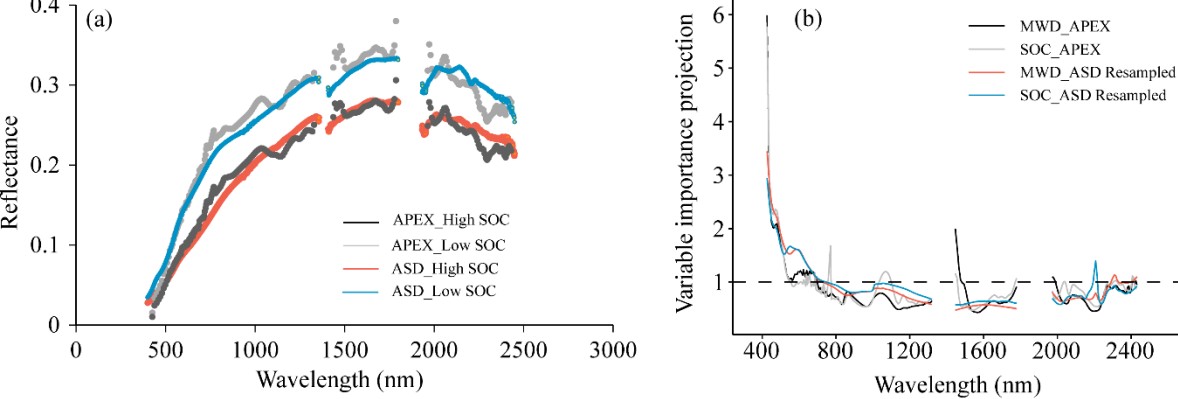

**Figure 4.** (**a**) Comparison between extracted APEX spectra and field ASD spectra at two locations with distinct SOC contents. (**b**) Variable importance projection (VIP) in partial least squares regression (PLSR) model for MWD and SOC using the extracted APEX spectra. The APEX curves (MWD_APEX and SOC_APEX) were compared to two other independent VIP curves generated using the laboratory ASD spectra (adapted from Shi et al. [29]), which are resampled to the same spectral resolution as the APEX spectra. Spectral wavelength with VIP value greater than one was considered significant for the PLSR model.

Examination into the importance of different spectral regions in the prediction models shows that similar spectral regions were involved in the prediction of both MWD and SOC (Figure 4b). Specifically, the most important wavelength for both variables was in the visible region between 400 nm and 600 nm, followed by the region around 750 nm and 1000 nm for SOC, and around 1450 for MWD. The two APEX-based VIP curves (i.e., MWD_APEX and SOC_APEX) were later compared to ASD-based (resampled to APEX spectral resolution) VIP curves (i.e., MWD_ASD Resampled and SOC_ASD

Resampled), which were generated using the laboratory spectra based on the same set of soil samples. The ASD-based PLSR model for MWD and SOC generated similar VIP curves that both displayed peaks in the visible region, with the exception that the SWIR (short-wave infrared) region around 2200−2400 nm was also important for the laboratory-based predictions. It is well-established that the visible region at 400−600 nm reflects the influence of SOC content [38]. This is in agreement with what was observed in the SOC prediction by the APEX spectra. Moreover, due to the close correlation between MWD and SOC, the visible region was also found to be important in the MWD prediction for both the laboratory and APEX models. A slight difference between the two models using different data sources was that the SWIR region was not significant in the APEX- prediction. This could be attributed to the low signal-to-noise ratio of this region in the APEX data, so that the spectral features in the SWIR region, which typically corresponds to the quantity of organic compounds like lignin and cellulose, are not well represented in the hyperspectral remote sensing images [23].

The calibrated and cross-validated PLSR model for SOC was further evaluated against an independent validation dataset using the predicted map as a reference for stratified random sampling (see Section 2.4.3 for details). Overall, good prediction accuracy in terms of RMSE and $R^2$ was achieved for the SOC model (Figure 5), although it was found that the model generally over-estimated the SOC values larger than 1.5%. Nonetheless, the model was capable of reproducing the general trend of SOC variation and had a similar level of RMSE and a better $R^2$ than the cross-validation results (Table 2). To avoid the time-consuming measurements of AS, the independent validation was only carried out against the SOC prediction. However, it is reasonable to assume a similar, if not better, performance for the AS prediction models, considering the cross-validation results of the AS models, the positive correlation between MWD and SOC, and the similar spectral regions that contributed to the predictions of both MWD and SOC. Hence, predicting AS by means of airborne hyperspectral remote sensing was proven to be effective, and it offers great potential to further investigate AS in a spatial context.

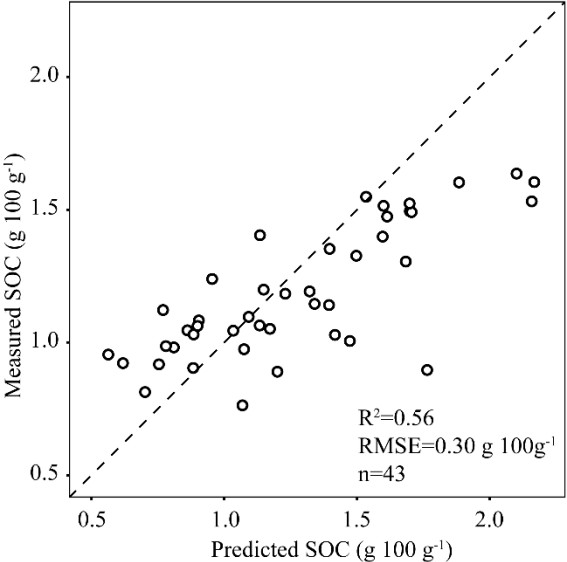

**Figure 5.** Scatterplot of measured versus predicted SOC using the independent validation dataset.

### 3.3. SOC and AS Mapping

The calibrated PLSR models for SOC and MWD were applied to the entire region covering 788 fields with more than 10 million pixels. Analysis on the statistical distribution of predicted SOC shows that the region is characterized by a low SOC content, with mean SOC concentration of 1.16 g C per 100 soil. Histogram analysis of MWD over the entire study region resulted in a normal distribution with a mean MWD of 0.36 mm (Figure 6), which is similar to the mean MWD of the calibration dataset (Table 1).

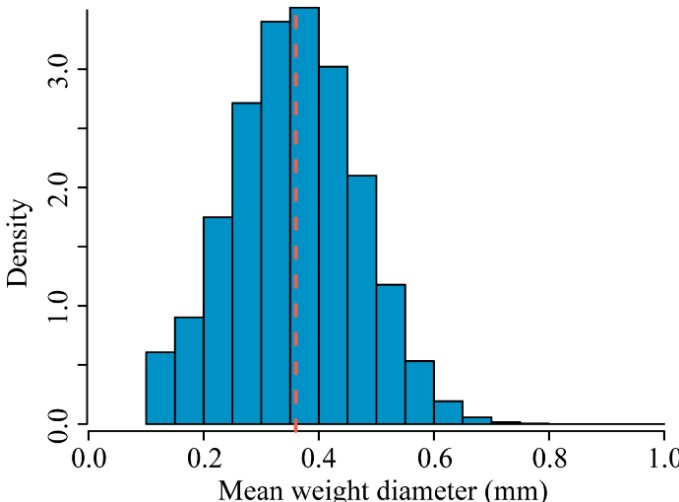

**Figure 6.** Histogram of the predicted soil mean weight diameter (MWD) across the entire study region. The vertical dashed line in red represents the mean MWD.

Adopting the classification criteria proposed by [8], who used 0.4 and 0.8 mm to separate the MWD into "highly unstable", "unstable" and "stable" classes, it can be seen that most of the soils in the study region are in "unstable" and "highly unstable" condition, posing great risks to soil degradation.

For the SOC map of a representative area, apart from the clear between-field variability of SOC concentration especially in the southwest corner of the selected area, the most striking feature was its in-field variability as highlighted by the dark "hotspots" of SOC concentration (as shown by the >2% SOC values in Figure 5) within the confined red rectangle in Figure 7a. Also visible on the APEX RGB image (Figure 7b), the dark spots were caused by the pre-industrial charcoal kilns abandoned more than 150 years ago [39]. The charcoal accumulation on the specific spots leads to higher SOC concentrations despite of the recent land use conversion to cropland, and the fact that the developed SOC prediction model was able to capture this spatial pattern demonstrates the potential of using high-resolution hyperspectral remote sensing techniques to investigate the small-scale variability of SOC across large areas. The MWD map displayed a similar pattern to that of the SOC map (Figure 7c). Between-field as well as in-field variability were observed and higher MWD values generally correspond to higher SOC values, confirming what was reported above on the positive correlation between MWD and SOC.

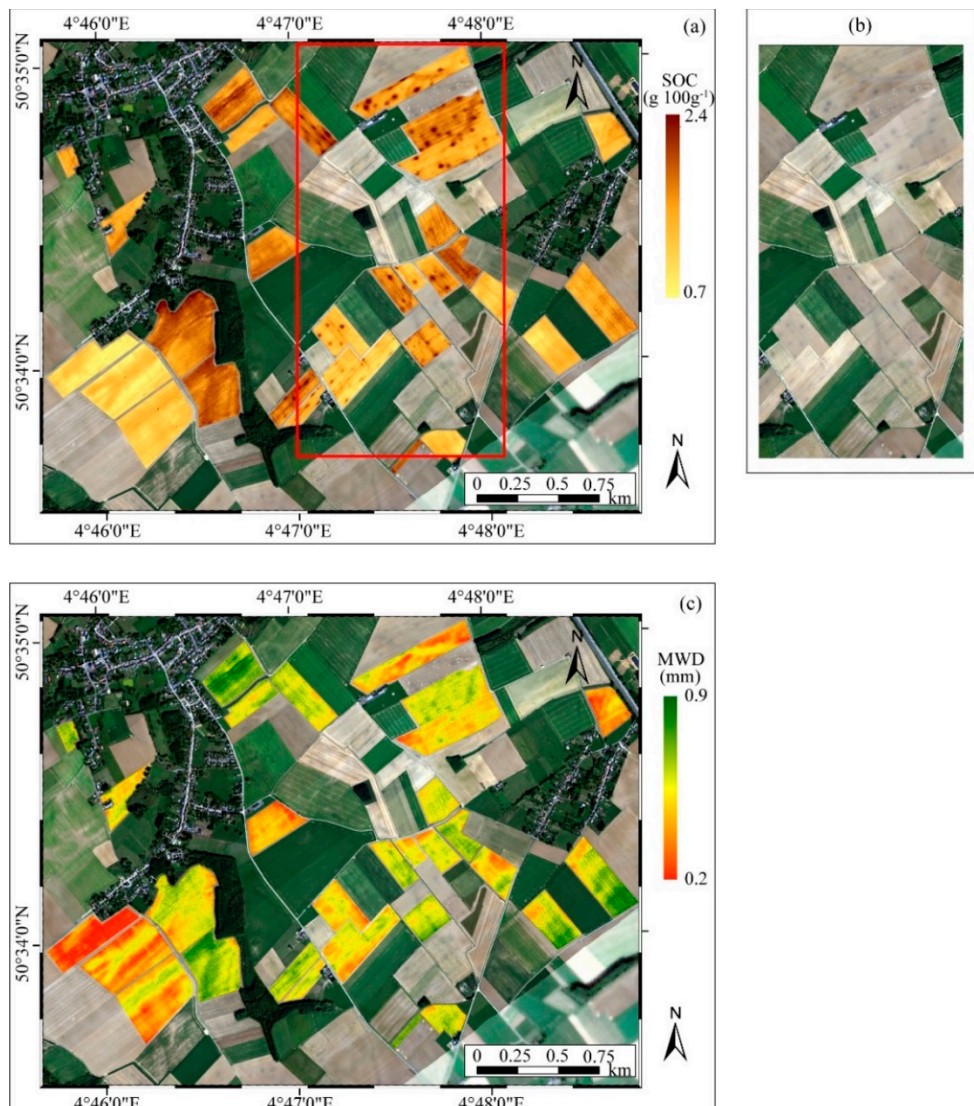

**Figure 7.** Zoom on a selected area of the (**a**) SOC map, (**b**) RGB image of a confined area shown by the red rectangle in (**a**), and (**c**) MWD map. SOC and MWD maps were produced by applying the PLSR models to the bare soil fields. The red square in the south-west corner of the map in Figure 1 gives the position of the selected area.

### 3.4. Spatial Variability of AS

Empirical variograms of MWD were computed at two different spatial scales, i.e., one covering all the bare fields of the study region to represent the macro-structure of spatial variation in AS, and two selected areas less than 5 km$^2$ to reveal any potential micro-structure. First of all, a clear macro-structure of MWD across the study region was observed, with the practical range of the fitted model at 1337 m (Figure 8). At a smaller scale however, contrasting spatial organizations were found in two selected areas (locations are shown in Figure 1). In SelectedArea1, the range of the fitted model was only 458.7 m, beyond which the spatial autocorrelation between points was estimated to be minimal. This distance (range) agrees well with the typical length (300–500 m) of an individual field in the region, indicating that field-specific agricultural practices and legacy effect of former land use or management may have caused large between-field variations [40]. Other possible mechanisms that could lead to discontinuous distribution of soil properties across fields include the presence of field boundaries (e.g. grass buffer strips, ditches, hedge rows) that intersect the lateral matter transfer driven by geomorphic processes [41]. It is also reasonable to speculate that topography did not play

a substantial role in determining the spatial pattern of AS, as SelectedArea1 is situated on a rather flat plateau where the elevation is not highly variable (Figure 1). In the second area, the range of the variogram model (1600 m) was at a similar level to that of the entire region but with a higher variability of MWD (semi-variance larger than 0.015). A variogram range beyond the scale of a single field implies that processes such as water erosion, which typically act at catchment scale, could be the driving force that shapes the spatial distribution of MWD. This is supported by the clear topographic pattern in SelectedArea2, which is characterized by sloping plateaus in the south and valley bottoms in the north (Figure 1). Previous investigations into the spatial variations in SOC in the same region also suggested that topography was an important determinant of SOC heterogeneity [42].

The contrasting spatial structures of MWD observed in different areas and across multiple scales imply that distinct driving factors were at play in determining the spatial distribution of soil properties. Almost all the studies that investigated the spatial variability of AS so far relied on point-based extensive soil sampling to investigate either the large-scale spatial variability based upon limited data [20] or the effects of landscape position and land use type on the variation in AS [3,43]. The sampling density in these studies was constrained by the resource-demanding laboratory analysis so that simultaneous consideration of both field- and regional-scale variability of AS was not possible. Here, we demonstrated the use of hyperspectral remote sensing images as an effective way to rapidly predict the large-scale spatial distribution of AS at a high resolution. This not only allows for analysis on the general patterns of soil properties to aid decision making in terms of agricultural management, but also provides detailed soil information in the context of precision agriculture and site-specific soil erosion assessment.

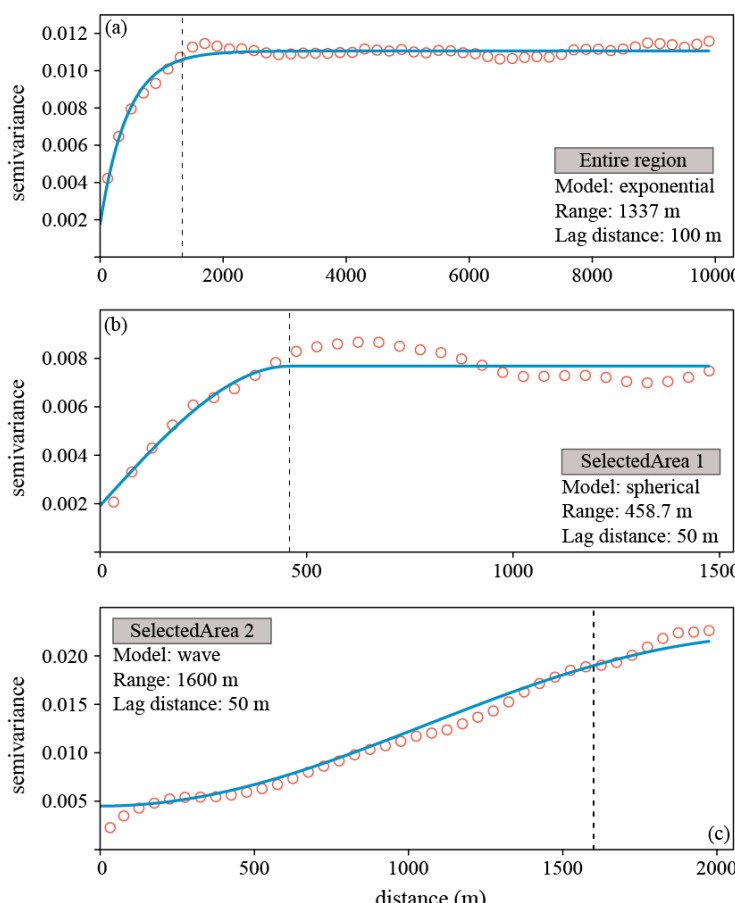

**Figure 8.** Empirical variograms (red circles) with fitted models (blue lines) for (**a**) the entire study region and (**b**,**c**) two selected representative areas. 300,000 pixels were randomly selected for the computation of empirical variogram for the entire region, while 100,000 pixels were randomly selected for each selected area.

The MWD map produced by hyperspectral imagery represents one of the first large-scale AS assessments, covering more than 700 agricultural fields. However, this genre of approaches relies on the successful extraction of bare soil fields, which are usually discretely located in space and only available during a short time window (when croplands are harvested and prepared for the next crop). To be able to produce spatially continuous soil maps, algorithms that assimilate multi-temporal composites are therefore needed to enable the mosaicking of bare soil pixels (free of disturbance from soil moisture and dry vegetation residue) across long time periods. In this context, the next generation of satellite-based hyperspectral imagers such as EnMap [44] and PRISMA [45] will provide more frequent hyperspectral images over large spatial scales, thus offering new opportunities to monitor soil degradation in space and time.

## 4. Conclusions

The capability of Airborne Prism Experiment (APEX) hyperspectral imagery to predict soil aggregate stability (AS) was tested in an agricultural region (ca. 230 km$^2$) of the Belgian loam belt. Partial least squares regression models were built using the extracted APEX spectra with three AS indexes (i.e., mean weight diameter (MWD), microaggregate and macroaggregate fractions) and soil organic carbon (SOC) as the response variables. We delineated a total of 788 bare soil fields from the images using predefined spectral indexes (NDVI and soil moisture index) that are representative of bare, dry soils. PLSR models were then applied to these fields to generate high resolution (2×2 m) maps for MWD and SOC. Satisfactory prediction models (RPD > 1.4, $R^2$ > 0.5) were achieved for all four variables, with the AS prediction models producing even better performance than the SOC model. Spectral regions that were known to be SOC predictors were also found to be responsible for the prediction of MWD due to their positive correlation. The predicted SOC and MWD maps proved the potential of hyperspectral remote sensing technique to capture both the fine-scale, in-field as well as between-field variability of these soil properties. Variogram analysis on MWD further revealed its contrasting spatial structures at different scales. Over the entire region, the spatial heterogeneity of AS was found to be driven by geomorphic processes like water erosion acting at catchment scale, while small scale investigations suggested that field-specific practices related to agricultural management could also lead to large variations in AS across neighboring fields.

**Author Contributions:** Conceptualization, P.S., F.C., B.v.W. and K.V.O.; methodology, P.S. and F.C.; validation, P.S. and B.v.W.; formal analysis, P.S.; investigation, P.S.; writing—original draft preparation, P.S.; writing—review and editing, P.S., F.C., B.v.W. and K.V.O.; supervision, B.v.W. and K.V.O.; project administration, P.S. and K.V.O.; funding acquisition, P.S. and K.V.O. All authors have read and agreed to the published version of the manuscript.

**Funding:** This research was funded by the Early postdoc. Mobility fellowship of Schweizerischer Nationalfonds zur Förderung der Wissenschaftlichen Forschung, grant number P2EZP2_178494. The APC was funded by National Natural Science Foundation of China, grant number 41807059. The APEX imagery was acquired within the STEREO III Belair program of the Belgian Federal Science Policy Office (BELSPO) and the support is gratefully acknowledged.

**Acknowledgments:** We thank VITO Remote Sensing Department for providing the APEX images. We are also grateful to Marco Bravin for his assistance with the sample collection and analyses.

**Conflicts of Interest:** The authors declare no conflict of interest.

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
