# Peer review of "Large-Scale, High-Resolution Mapping of Soil Aggregate Stability in Croplands Using APEX Hyperspectral Imagery"

_remotesensing, doi:10.3390/rs12040666_

Round 1

Reviewer 1 Report

The paper is based on the same experimental dataset of an already published study by the same authors, indicated in the reference at [29]. The analysis is conducted by using a different approach from that used in [29], but with similar conclusions.

The paper is clear in most sections, however it could be more readable respect to the following points:

Sect.2.3: explain why soil sampling was not carried out in a period closer to the flight acquisition rather than one month later. The time lag is responsible for the reduced dimensionality of the ground dataset used in the calibration? Only 49 samples out of the 83 collected from bare soil fields were used (line 243) Sect.2.4.3: which spectral data were used for the independent validation? Sect.2.5: is the investigation of the semivariogram finalised to mapping by means of Kriging? This analysis seems to be out of the main context of the paper.

Author Response

Dear reviewer,

Thank you very much for your review of our manuscript. We have addressed the issues raised by you in the following.

Sect.2.3: explain why soil sampling was not carried out in a period closer to the flight acquisition rather than one month later. The time lag is responsible for the reduced dimensionality of the ground dataset used in the calibration? Only 49 samples out of the 83 collected from bare soil fields were used (line 243)

The one-month time lag after the APEX flights was due to logistic reasons and changing weather conditions during that period. We did try to complete the soil sampling as soon as we could.

The primary reasons why we only managed to select 49 samples for calibration were: 1) a portion of the 83 samples was not in the APEX flight zone, because the APEX flight campaign in 2018 covered a slightly different region than the one in 2015, which we used as a reference to choose our sampling locations; and 2) we used a bare soil detection criteria that excluded the samples that were influenced by soil moisture, and this led to further decrease in the calibration samples.

Sect.2.4.3: which spectral data were used for the independent validation?

We carried out independent validation to evaluate the performance of the developed prediction model for soil organic carbon (SOC). The validation was to evaluate the predicted SOC values (using the model based on extracted APEX spectra) against the measured SOC values. No spectral data was involved in this procedure.

Sect.2.5: is the investigation of the semivariogram finalised to mapping by means of Kriging? This analysis seems to be out of the main context of the paper.

The use of semivariogram was to analyze the spatial structure (indicated by the range of the semivariogram) of aggregate stability (AS) across different spatial scales, so as to highlight the point that our approach was able to produce accurate and high resolution AS data, thus enabling the assessment of the spatial variability of AS at field, catchment and regional scales. No kriging was performed.

Reviewer 2 Report

Dear authors, the paper presents an interesting application of multispectral remote sensing data in digital soil mapping. Nice work and results! Please see my comments in the .pdf file for some minor corrections.

Author Response

Dear reviewer,

Thank you very much for your review of our manuscript and your recognition of the quality of our work.

Following your suggestions, we have corrected some sentences. For the one in Line 214, we felt that it was approriate to start the paragraph by mentioning the purpose of independent validation was to further assess the robustness of the model, we therefore kept it unchanged.

We hope that the manuscript has now been corrected to your satisfaction.

Best wishes,

Pu Shi

Reviewer 3 Report

Figure 1: Please show different symbols for data collected at different acquisition dates.  A missing information in this manuscript is “what constitutes the calibration and validation data and when those were collected in relation to when the hyperspectral image was taken.”  The axes for this figure should be changed to latitude and longitude!

Line 125, 126, 140: When, which, and how many ground data locations coincided with hyperspectral data acquisition?

Lines 214-219: For these 43 locations, hyperspectral data were also collected in October 2019?

Line 215-216: This is not clear! “SOC as a proxy for AS estimation” through hyperspectral data should be established if we were to use SOC as a method of testing hyperspectral approach for AS, is not it? So, providing SOC validation discussion takes a reader away from the main theme of aggregate stability assessment!

Line 231: What does “representative” mean here? Does it mean that this strip was completely bare soil? If no, then how MWD data for each pixel was estimated?

Line 243: With the data available for 83 locations, the authors should have considered an unmixing approach. If not, at least, the method developed by Dutta and Kumar (https://ieeexplore.ieee.org/abstract/document/8579521) may have been considered.

Line 254 and Figure 3: If we draw a straight line through the whole dataset (without splitting it into two clusters), we should also see a reasonable fit!! In that case, what is the need for splitting SOC into two categories? Yes, having a corresponding critical limit for AS is good, but I am not sure if this dataset (Figure 1) justifies the presence of such a critical limit.

Figure 5: Similar plot for AS should be included and some kind of upper and lower bound for corresponding RMSE (may be through bootstrapping) should also be included in the manuscript.

Line 350-351: Please include measured SOC content data for the hot spots considering that these occupy only a small fraction of the total study area.

Line 383: How such a conclusion is drawn? There could be many reasons behind the variogram result!!

Line 366: How MWD data were obtained for entire study region (~ >200 km2).

Reviewer 4 Report

Thank you for this hard work and very interesting manuscript. I have only several minor recommendations and questions. 

Abstract:

": Investigations into the spatial dynamics of soil aggregate stability (AS) are urgently 15 needed to better target areas that underwent soil degradation."

This sentence is not clear, should be revised.

"detailed information on its 17 spatial structure across scales are scarce"

Once again it is hard to understand, why scales are important to AS? Description must be given prior to this general and a bit foggy statement.

"high-resolution AS map at 19 regional scale."

Once again, this need is not clear. AS is a local factor, why would it be interesting on a regional scale?

"hyperspectral images covering an area of 230 km2 20 in the Belgian Loam Belt were used together with a local soil dataset."

The image is on surface only, however, " a local soil dataset" is a pedon, therefore, more detailed info is needed alredy in the abstract on the method. As next, the reader is overloaded by empirical-statistical models that have no physical justification. Thus, it shows loose and simplified approach.

" the fine-scale as well as the between-field variabilities of soil 27 properties"

Yet the model was developed on a regional scale. This is one big conflict.

" Variogram analysis on the spatial structure of MWD showed a clear spatial organization 28 at the catchment scale (range: 1.3 km) that is possibly driven by erosion-induced soil redistribution 29 processes"

Without any justifications, this statement is too ambitious, must be revised.

" a promising technique to investigate the spatial variability of AS across multiple 32 scales."

Again this need is not clear.

Introduction

" soil erodibility to reflect a soil’s inherent resistance to external 57 erosive forces"

This is relevant for pedon in soil profile, how is it applicable in this study?

" key soil properties, such as SOC and soil texture"

On surface only. This must be clearly stated.

Methods

" silt loam soils on a rolling topography"

This is super general description. Must be revised.

" MODTRAN4 radiative transfer model following the algorithms given in"

Did you correct the BRDF. This might be a strong effect on the surface reflection, especially for bare soil plans, especially on a rolling topography.

" Then, the post-processed images were 135 resampled to a spatial resolution of 2×2 m and projected to WGS 84/UTM zone 31N."

Why?

" 83 topsoil"

Why this number of sample? Why this samples were not measured by spectrometer before collection and in the laboratory? This could be your spectral ground-truth validation

" SMI values above this 178 threshold were assumed not to be disturbed by the noise caused by variation in soil moisture"

Must be added – on top soil only

" the “bilinear” method from the “extract” function provided by the R package “raster"

Why? it is just an extra interpolation on spectral data.

Results and conclusions are well presented and detailed described. 

Reviewer 5 Report

The authors present a research paper on mapping soil aggregate stability using APEX data. The paper is well-written and interesting to read. In my opinion it can be accepted as is.

One minor thing that might be changed is the axes in figure 5. It is quite unusual that high SOC values are over-estimated (please check if your plot is correct), and if the axes were switched, the over-estimation would be more obvious.

Author Response

Dear reviewer,

Thanks a lot for your time to review our manuscript and for your recognition of our work. 

We have double-checked Figure 5, and we confirm that thre figure is correct and there does seem to be an over-estimation of the high SOC values. Future work to include more calibration samples in the high SOC range might increase the model performance. 

Thanks again, and best wishes,

Pu Shi

Round 2

Reviewer 3 Report

Thank you for incorporating a few of my suggestions! Good luck with the publication!